# Effective Active Ingredients Obtained through Biotechnology

**Claudia Zappelli [1], Ani Barbulova [2], Fabio Apone [1,2,*] and Gabriella Colucci [1,2]**

[1]  VitaLab srl, via B. Brin 69, Naples-80142, Italy; claudiazappelli@vitalabactive.com (C.Z.);
     gcolucci@arterrabio.it (G.C.)
[2]  Arterra Bioscience srl, via B. Brin 69, Naples-80142, Italy; ani@arterrabio.it
[*]  Correspondence: fapone@arterrabio.it; Tel.: +39-81-6584411

**Abstract:** The history of cosmetics develops in parallel to the history of man, associated with fishing, hunting, and superstition in the beginning, and later with medicine and pharmacy. Over the ages, together with human progress, cosmetics have changed continuously and nowadays the cosmetic market is global and highly competitive, where terms such as quality, efficacy and safety are essential. Consumers' demands are extremely sophisticated, and thus scientific research and product development have become vital to meet them. Moreover, consumers are aware about environmental and sustainability issues, and thus not harming the environment represents a key consideration when developing a new cosmetic ingredient. The latest tendencies of cosmetics are based on advanced research into how to interfere with skin cell aging: research includes the use of biotechnology-derived ingredients and the analysis of their effects on the biology of the cells, in terms of gene regulation, protein expression and enzymatic activity measures. In this review, we will provide some examples of cosmetic active ingredients developed through biotechnological systems, whose activity on the skin has been scientifically proved through in vitro and clinical studies.

**Keywords:** cosmeceuticals; biotechnology; anti-aging; efficacy

## 1. Introduction

Scientists have always been fascinated by the biological, social and medical implications of beauty. However, the importance of scientific studies as applied to cosmetic science has undergone a profound revolution guided by an increasing knowledge of skin physiology. Cosmetic brands as well as consumers have progressively interrogated themselves about the real efficacy of cosmetic ingredients on the skin. Before the stringent guidelines for the evaluation of cosmetics, personal care products were often designed as mere instruments of adornment, where no real demonstrated efficacy was prominent for many of the cosmetics products. Traditionally, cosmetics have been considered preparations, such as lotions or creams, designed to enhance personal appearance by direct application on to the skin and offering "hope in a jar".

Fortunately, scientific research has made a huge step forwards and all the cosmetic ingredient producers are now aware that science, in particular cell and molecular biology techniques, not only can help in understanding what the real efficacy of cosmetic products is, but represents a key tool to confirm whether biologically active ingredients intended for production are safe for consumers. The jar does not represent only a "hope", but can effectively ameliorate several skin conditions.

This change of direction was also driven by the shift of interest in the final consumers: from attention for the most colorful and seductive package to the curiosity of knowing what the latest scientific breakthrough was about. Active ingredients as well as final products with scientifically demonstrated efficacy on the skin are traditionally defined as "cosmeceuticals", and they have now

become the key drivers in the cosmetic industry. It was Albert Kligman, an American dermatologist, who in the last quarter of the 20th century opened what we could define "a transition phase" for the cosmetics industry, coining the term "cosmeceutical" to describe a new form of cosmetic combining cosmetic and pharmaceutical properties meeting consumers demands for high efficacy [1]. Cosmeceuticals are not simply traditional cosmetics, but part of a skin care routine aimed at creating physiological changes in skin cells to make them appear healthier and younger. Treatment of the early signs of sun damage is quite clearly preventive medicine, and so one really needs "medicines" to deal with the problem. This is where cosmeceuticals come into the picture. We need active cosmetic ingredients that have a real therapeutic effect on the skin. The word cosmeceutical conveys the meaning of a cosmetic product having a pharmaceutical effect analogue to that of a topical drug, and this concept introduced a revolution in skin care.

In spite of the extensive use of the word "cosmeceutical" in the personal care segment, the term is not recognized by the Federal Food, Drug, and Cosmetic Act (FD&C Act) to date, and presently, only Korea and Japan have legislation that distinguishes three categories: cosmetic products, functional (quasi-drug) cosmetics, and drugs. That said, in the present article we will define the substances used in a cosmetic formula to achieve proven local biologic effects and which are obtained through biotechnological techniques as "active ingredients" or "cosmeceuticals".

Biotechnology has impacted cosmetics in many ways. Cosmetic companies use biotechnology both to discover, develop, and produce components of cosmetic formulas, and to evaluate the activity of these components on the skin, in particular how they impact the changes associated with aging. Consumer safety and testing reproducibility must be guaranteed by cosmetic ingredient producers, as they undergo extremely strict manufacturing and production environment inspections. Moreover, both the industry and the consumers are now aware that bio-sustainable ingredients are simply better for the environment. Fewer resources such as water, soil and/or electricity must be exploited; plus, the ingredients must not be exposed to pesticides and pollutants, providing higher safety levels for the final consumers.

Over the past decade, cosmetic companies have invested heavily in molecular, genomic and proteomic research into what causes skin cells to age, with the hope of pinpointing ways to interfere with that process. When evaluating the success of both cosmetic and cosmeceutical products, we must consider the appropriate integration of skin structure and functional aspects with the nature of the formulation, its efficacy as defined by the goal of the product, and its safety. According to their functions or effects, anti-aging active ingredients can be roughly classified into three main categories: moisturizing, antioxidant and extra-cellular matrix boosters. Active ingredients can differ in their manufacturing origin as well as in their biological mechanism of action. In this review, we want to focus on some active ingredients developed through biotechnological systems, whose activity on the skin has been scientifically demonstrated through in vitro and clinical studies.

## 1.1. Antioxidant Active Ingredients

The biological processes of aging, the longevity of multicellular organisms, and the role of the genome have been major focuses of scientific research for many decades. Various theories of aging have been suggested and supported by scientific evidence. The oxidative stress theory of aging remains the most popular explanation, suggesting that progressive accumulation of oxidative DNA damage is a contributory factor to the aging process. As we age, our susceptibility to reactive oxygen species (ROS) increases because our ability to synthesize antioxidants decreases, so the inherent epidermal antioxidant defense system declines and skin damage increases. In their review article, Dreher et al. underline that "regular application of skincare products containing antioxidants may be of the utmost benefit in efficiently protecting our skin against exogenous oxidative stressors occurring during daily life" [2]. Furthermore, sun-screening agents may also benefit from a combination with antioxidants, resulting in increased safety and efficacy of such photoprotective products [2].

Vitamin E, vitamin C, superoxide dismutase (SOD), coenzyme Q10, zinc sulfate, ferulic acid, idebenone, polyphenols and carotenoids are examples of free-radical-scavenging molecules, which have been successfully employed in cosmetic products for a long time. Biotechnology represents a good alternative tool for developing active ingredients from natural sources that are able to strongly counteract oxidative damages, therefore slowing down the aging process. Studies conducted on plant cell wall membrane glycoproteins revealed their enhanced expression in response to tissue damage and other types of physical and chemical stresses [3,4]. These studies highlighted that cell wall integral glycoproteins are critical for the perception of external stimuli and activation of downstream defense response signaling in plants [5]. Researchers developed a suitable method for the extraction, the purification and the processing of these glycoproteins from plant cell walls, in order to obtain a mixture of small peptides, rich in glycine, proline and hydroxyproline, and sugars, which could be potentially effective in activating defense response mechanisms in human skin cells as well [6]. In vitro studies showed that the mixture of peptides and sugars, obtained from the plant cell walls, was able to activate specific signaling pathways leading to the up-regulation of anti-aging genes, making cells more resistant to stress factors and stimulating the synthesis of skin extracellular matrix component pathways [6]. Clinical data, obtained from human volunteers through double-blind tests measuring wrinkles and roughness using the Skin-Visiometer®, confirmed the in vitro results and indicated that the treatment with the peptide/sugar mixture effectively reduced the face line and wrinkle depth by 25% after 30 days of application [7].

Another active ingredient developed from plant cell cultures, which showed interesting antioxidant proprieties, was verbascoside, a phenylpropanoid glycoside [8]. Besides its oxidant protective action, this compound was characterized for its anti-inflammatory activity, inducing a dose-dependent decrease of the expression of the pro-inflammatory chemokine IL-8 on primary cultures of human keratinocytes stimulated by TNF-$\alpha$, a cytokine, involved in the acute phase reaction in the systemic inflammation [9]. Verbascoside was obtained from *Buddleia davidii* meristematic cells, obtained in turn using a sustainable biotechnological platform, which employs an in vitro plant cell culture technology, and thus is an interesting example of how plant cell cultures can be used as biofactories to produce specific compounds of interest having good therapeutic potential.

Besides using biological systems, many compounds can be more conveniently produced by chemical synthesis, whether the process meets the requirements of high production yield and sustainable costs. This was the case of the biomimetic peptide ethylbisiminomethylguaiacol manganese chloride (EUK-134™), characterized as a natural effector in skin physiology regulation [10]. When tested on human keratinocytes, EUK-134™ acted as an antioxidant and anti-inflammatory agent, scavenging superoxide free radicals, eliminating hydrogen peroxide and reducing P53 expression. P53 is well known to accumulate following various types of environmental stresses, including DNA-damaging agents, decreased oxygen, and heat shock, and redox stress [11]. Declercq et al. [12] recently supported the concept that the synthetic SOD/catalase mimetic EUK-134 might be able to compensate for seasonal deficiencies in the antioxidant defense capacity at the skin surface, thereby contributing to an optimal protection of the skin against the accumulation of oxidative damage and confirming previously published results from Decraene et al. about the protective effects EUK-134 on human keratinocytes [12].

### 1.2. Active Ingredients for Moisturization

Moisturizing/hydration is an essential part of skin care. The very top layer of the skin is known as the stratum corneum, and it is made up of dead skin cells in a matrix of lipids. Twenty percent of the stratum corneum is composed of lipids, and the major ones are cholesterol, fatty acids, and ceramides. In general, unsaturated fatty acids reinforce the skin's barrier function, prevent water loss through the epidermis, and provide structural integrity to the skin against damage by external influences. They also help to soften and smooth the skin by promoting desquamation [13,14]. Certain ingredients have been shown to help moisturize the skin. Taking into account that ceramides are

comprised of 50% of the lipid content in the stratum corneum, maintaining ceramide levels is crucial in sustaining hydration and barrier function of the skin. For many years plant seed oils have been used as a moisturizer due to their high fatty acid content [15].

Examples of compounds with skin moisturizing activity come from the study of sugar biology. The synthesized xylitylglucoside, a sugar derivative of two plant sugars, xylitol and glucose, stimulated the synthesis of proteins, enzymes (different types of keratin, loricrin, transglutaminase) and ceramides essential to the function of the skin barrier. Moreover, the synthesis of dermal macromolecules, such as hyaluronic acid and chondroitin sulphate, able to "trap" water was also increased in fibroblast culture. Clinical tests demonstrated that the epidermal water content was improved on the first day of treatment (demonstrated in vivo on 25 volunteers) and that trans-epidermal water loss was reduced after one month of treatment (demonstrated in vivo on 25 volunteers) [16].

Plant cell cultures may represent an interesting source of active ingredients specifically aimed at improving the moisturizing potential as well. The example comes from *Rubus idaeus* cell cultures, which were used to develop an oil-soluble extract particularly rich in essential fatty acids [17]. The extract induced, in cultured keratinocytes and fibroblasts, the expression of the most important genes involved in skin hydration and moisturization, such as aquaporin 3, filaggrin, involucrin and hyaluronic acid synthase. Clinical tests confirmed the good moisturizing properties of the natural extract and indicated a significant hydrating effect on the skin even after 7 h from the application (short term) and it kept the skin hydrated over a longer time period (+16% after 14 days and +19% after 28 days).

### 1.3. Cell Repairing and Matrix Booster Ingredients

The extracellular tissue matrix (ECM) of the skin is a complex aggregate of distinct collagenous and non-collagenous components. This connective tissue is produced, organized, and maintained by dermal fibroblasts. Optimal quantities and delicate interactions of the various components are necessary to maintain normal physiologic properties of the skin. During aging, ECM components undergo progressive loss and fragmentation, leading to thin and structurally weakened skin. For these reasons, components found within ECM have emerged as an essential target for achieving efficacy in active ingredients. Collagen-, elastin- and ECM-degrading enzymes are the main categories of target genes studied in a wide variety of raw materials.

"Matrikines" are peptide fragments whose sequence is generally less than or equal to 20 amino acids, derived from matrix proteolysis during cutaneous wound cleaning prior to healing. Generated peptides act as autocrine and paracrine messengers able to regulate, upstream, the sequence of events necessary for satisfactory wound healing, exerting positive feedback on the process of connective tissue renewal and cell proliferation, and are formed in larger quantities [18]. A peptide of particular interest is lysine-threonine-threonine-lysine-serine (KTTKS), which is a fragment of pro-collagen I. Katayama et al. previously reported that this pentapeptide positively stimulated dermal matrix production in fibroblast culture and demonstrated that the significant stimulation of type I collagen, type III collagen, and fibronectin production occurred in a dose- and time-dependent manner with no effect on total protein synthesis or on the ratio of secreted proteins to cell-associated proteins [19].

However, due to the ionic nature of the peptide, the skin penetration and consequence efficacy might be limited. In order to improve the penetration through the lipidic structures of the skin, the peptide of five amino acids was recently linked to a 16-carbon chain of the molecule, forming palmitoyl-KTTKS (pal-KTTKS) [20,21]. The authors demonstrated improved delivery across the skin compared to the non-conjugated pentapeptide form, and also reported facial skin improvement effects in a 12 week clinical test. The mechanism by which pal-KTTKS operates is likely via stimulation of the dermal matrix production as previously shown for KTTKS. Worth mentioning is also Bombesin, a 14-amino-acid neuropeptide shown to act on tissue regeneration and wound healing of the skin in an in vitro experimental model [22]. Small peptides have gained increasing interest thanks to their proven efficacy and versatility. An example is the tripeptide composed of the amino acids glycine, histidine

and lysine (Gly-His-Lys) which stimulates collagen synthesis in human skin fibroblasts, as found by Maquart et al. [23].

Peptide engineering has made enormous progress in the last decades thanks to biotechnology and results obtained after isolation, followed by chemical, biochemical and biological characterization of several peptides being published [24,25]. The identification of suitable modification that allows peptides to be targeted into the various skin layers contributes to employing peptide sequences for cosmetic purposes a viable strategy.

## 2. Conclusions

Until recent times, cosmetics were not broadly associated with science because the final consumers were probably scared of and certainly uninterested in the possible biological effects of the products once applied on their skin. Since then many things have changed and the common perception of cosmetics being unscientific changed drastically when companies started heavily investing in effectiveness and claim substantiation.

Medical science and biotechnology have played increasingly greater roles in the beauty industry due to growing consumer demand for advanced cosmetic formulations and efficacious ingredients. Cosmeceuticals are the manifestation of a convergence of science, medicine and beauty, and have revolutionized the world of skin care. By understanding the basic scientific mechanism of how cosmeceuticals work on the skin, one can design more specific products, addressing a wide range of functions that the end user can rely on.

Using uncharacterized plant extracts in skin care creams, most of the time containing unknown compounds, is no longer appealing nor desirable. Currently used plant extracts require a high definition and availability in standard concentrations with scientifically proven effects. Claim substantiation holds efficacious products to high standards of scientific validation. Thus, there is a need for clearly understanding that the ingredient is really effective on the skin, that it has a defined mechanism of action, and that it produces specific clinical effects with continued topical use.

The latest tendencies of cosmetics are based on advanced research that includes the use of biotechnology-derived ingredients, genetic profiling for individual skin care or nutritional regimes, stem cell–based products and therapies to regenerate aging tissues, or cell and tissue engineering for cosmetic purposes. The "hybrid" skin care products that mix the boundaries between cosmetics, pharmaceutics and nutraceutics, together with the growing number of claims for natural/organic products, demand further regulation and common standards [26,27].

The main regulatory frameworks governing the cosmetic industry date back to 1938 in the United States (US), and 38 years later in Europe (EU). Since then, both the US [28] and EU [29] cosmetic legislations have inspired the regulatory framework of a number of countries, working toward the harmonization of cosmetic ruling. Even though the requirements for the efficacy of cosmetic products have been implemented to adapt them to the state of the art, no clear guidelines for efficacy testing on cosmetic products exist [30].

According to The New EC Cosmetics Regulation 1223/2009, cosmetic products are required to be effective when used by consumers under normal, labeled or foreseeable conditions of use. The European regulation CE 655/2013 states that claims for cosmetic products, whether explicit or implicit, shall be supported by adequate and verifiable evidence regardless of the types of evidential support used to substantiate them, including appropriate expert assessments.

The cosmetic claims should be an integral part of product development and design. According to the Colipa Guidelines for Efficacy Evaluation of Cosmetic Products [31], the main evaluation methodologies, which provide an appropriate and effective tool to assess the product efficacy, at the same time facilitating the innovation and the competition, are: (i) the sensorial approach (sight, touch, olfaction) by consumers themselves or experts; and (ii) the instrumental approach, which favors specific criteria measured using in vivo, ex vivo or in vitro models, which do not reproduce normal conditions of the use of products, but allow objective analysis of specific activities.

One must not forget, however, to take into account all aspects in this field:

- The active ingredients so developed must be proven to be safe for the consumer; extensive toxicity testing in vitro and in vivo needs to be performed to exclude possible risks for human safety.
- Medical application and curative action on diseased skin must not be claimed.
- Conformity to all regulatory requirements and respect of all existing intellectual property rights is mandatory.
- There must be compatibility with a wide range of cosmetic formulations such as creams, lotions, gels, shampoos, and balms. The reasonable cost of production and commercialization should guarantee fairness to both parties involved in the transaction.

Biotechnology and pharmaceutical companies are making huge investments in new technologies to drive product innovation. In the last decade we greatly increased investment in scientific research and new technologies, driven by the assumption that "defeating" skin aging with beauty products will become increasingly possible. Biomimetics, 3D bioprinting, and plant tissue culture technologies will all be areas of research of the beauty industry in the future. All of these developments should ultimately benefit consumers by resulting in the next generation of safer and more efficacious products.

In this contribution, we have attempted to show how research progress and specific interests of scientists, cosmetic marketers, dermatologists, and consumers alike potentially drive the discovery of functional ingredients with proven efficacy.

**Conflicts of Interest:** The authors declare no conflict of interest.

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
