# Peer review of "Effective Active Ingredients Obtained through Biotechnology"

_cosmetics, doi:10.3390/cosmetics3040039_

Round 1

Reviewer 1 Report

Issues found throughout document:

Errors in capital letter usage;

Inconsistent spelling of aging vs. ageing (both are generally accepted, but one form should be used throughout)

Use of the term “proven”: recommend replacing with “demonstrated” or “shown”

Line-item issues:

12: inappropriate capital letter usage in “Cosmetics”

17: inappropriate capital letter usage in “Cosmetics”

37: recommend rephrasing: “scientific research has made huge steps forward”

38: inappropriate capital letter usage in “Science”, “Cell and Molecular Biology”

40: recommend rephrasing “…but represents a key tool for determining whether biologically active ingredients intended for production are safe for consumers  as written, this sentence was unclear.

44: recommend replacing “proven” with “demonstrated” or “shown”

50: inappropriate capital letter usage in “Cosmetics”

57-58: remove “it is mandatory to say that”  this does not help clarify the sentence and could lead to misunderstandings.  The FDA does not mandate, regulate, or recognize cosmeceuticals in the US marketplace.

63: inappropriate capital letter usage in “Cosmetics” and “Biotechnology”

69: recommend replacing “Very few…” with “Fewer…” I think the intent here is to reduce environmental impact.  We may not be able to get it down to a level of “very few”.

70: “guarantying the highest safety level” should be rephrased.  Recommend “providing higher safety levels.”  We can not guarantee product safety or efficacy under all circumstances but do want to optimize safety and efficacy.

78: “Although active…” remove the term “Although” and simply rephrase to “Active ingredients can differ…”

81: recommend rephrasing: “has been scientifically demonstrated

84: use of ageing vs aging.

85: “…genome have been major focuses” recommend removing the term “one” as it is grammatically improper here.

86: use of ageing vs aging.  Recommend rephrasing 2nd sentence: “The oxidative stress theory of aging” (also, inappropriate use of capital letters in this phrase)

88: use of aging vs ageing

90: “…defence system gets damaged and increasingly less resolving” this sentence is unclear.  Recommend rephrasing “…defence system declines and skin damage increases.”

93: Reference #2 is a review article, not a primary source.  This report does not contain the original study demonstrating these results.  Either rephrase this sentence to indicate that it is a review or update the citation.

95: “…photoprotective products.” Provide citation.

100: “…oxidative damage”, ageing vs aging

104-107: cite this sentence about the glycoproteins, even if it is the same reference #3 as used in the subsequent sentence, or otherwise indicate that this is part of the same study.  I wanted to know where this data came from.

111: Clinical data is not provided in reference #3.  This sentence dealing with the Skin-Visiometer® also requires citation.

118-119: Significant reference issue.  Reference #5 was performed in lung lymphoblasts, not keratinocytes.  This study did not address IL-8 or TNF-α.

130: “reducing P53 expression which…” this sentence is somewhat unclear.  Recommend breaking it into two sentences: “reducing p53 expression.  P53 is well-known to accumulate…[7]” this also resolves a reference ambiguity.

136: “protective effects of EUK-134 on human keratinocytes” citation needed.

148: inappropriate use of capital letters in “Sugar Biology”

154: “improved on the first day of treatment” and (“in vivo proven” should be changed to “in vivo demonstrated”)

155: proven should be changed to demonstrated

172: inappropriate capital letter usage in Elastin; needs extra term for clarity: “Collagen, elastin, and ECM-degrading enzymes…”

176: “Generated peptides act” subject/verb agreement

192: inappropriate capital letter usage in “Biotechnology”

198: various grammatical.  “Until a decade ago cosmetics were not broadly associated with science”

199: “…uninterested in what the possible…” recommend removing the term what: “…uninterested in the possible…”

201: inappropriate use of capital letters in “Cosmetics” and “Companies”.  Need to rephrase “allergic to science”; recommend “unscientific”

203: inappropriate use of capital letters in “Medical Science and Biotechnology”

204: inappropriate use of apostrophe: “…due to growing consumer demand”

205:  inappropriate use of capital letters in “Science, Medicine, and beauty”

206: “By understanding the basic scientific mechanism” (singular/plural issue)

210: “…is not anymore appealing nor desiring” recommend rephrasing for clarity: “…is no longer appealing nor desirable.”

212: recommend replacing “higher standards” with “high standards”.  Higher is comparative: higher than what?

215: inappropriate use of capital letter in “Cosmetics”

216: inappropriate use of capital letter in “Biotechnology”

218-219: inappropriate use of capital letters in “Cosmetics, Pharmaceutics, and Nutraceutics”

224: recommend rephrasing “Even though the requirements for the efficacy…” the original phrasing is unclear.

240: recommend removing term “perfectly” as we can not make a product 100% completely safe.

241: recommend removing term “all” from “all possible risks”—again, we can not ensure complete safety.

248: “fairness to both parties”

250: “…we assisted to peaks…”  recommend rephrasing to “we greatly increased investment in…”  we did not climb a mountain (ascend vs. assist)

Ref 14: citation issue.  This article was originally published in Personal Care Magazine, Jan 2004 pp 9-14, NOT on ResearchGate! 

Author Response

We really appreciated the referee’s suggestions, they were all useful to improve the quality of the manuscript. All referee’s comments have been accepted and revised in the text.

All changes were highlighted in the “rebuttal note” version. Changes in the reference list and numbering in the new text were not highlighted to simplify the reading. Please find the answer to each comment below in red.

Reviewer 1:

- Errors in capital letter usage;

- Inconsistent spelling of aging vs. ageing (both are generally accepted, but one form should be used throughout)

- Use of the term “proven”: recommend replacing with “demonstrated” or “shown”.

These corrections were made.

Line-item issues:

12: inappropriate capital letter usage in “Cosmetics”

17: inappropriate capital letter usage in “Cosmetics”

37: recommend rephrasing: “scientific research has made huge steps forward”

38: inappropriate capital letter usage in “Science”, “Cell and Molecular Biology”

40: recommend rephrasing “…but represents a key tool for determining whether biologically active ingredients intended for production are safe for consumers”  as written, this sentence was unclear.

44: recommend replacing “proven” with “demonstrated” or “shown”

50: inappropriate capital letter usage in “Cosmetics”

57-58: remove “it is mandatory to say that”  this does not help clarify the sentence and could lead to misunderstandings.  The FDA does not mandate, regulate, or recognize cosmeceuticals in the US marketplace.

63: inappropriate capital letter usage in “Cosmetics” and “Biotechnology”

69: recommend replacing “Very few…” with “Fewer…” I think the intent here is to reduce environmental impact.  We may not be able to get it down to a level of “very few”.

70: “guarantying the highest safety level” should be rephrased.  Recommend “providing higher safety levels.”  We can not guarantee product safety or efficacy under all circumstances but do want to optimize safety and efficacy.

78: “Although active…” remove the term “Although” and simply rephrase to “Active ingredients can differ…”

81: recommend rephrasing: “has been scientifically demonstrated”

84: use of ageing vs aging.

85: “…genome have been major focuses” recommend removing the term “one” as it is grammatically improper here.

86: use of ageing vs aging.  Recommend rephrasing 2nd sentence: “The oxidative stress theory of aging” (also, inappropriate use of capital letters in this phrase)

88: use of aging vs ageing

90: “…defence system gets damaged and increasingly less resolving” this sentence is unclear.  Recommend rephrasing “…defence system declines and skin damage increases.”

93: Reference #2 is a review article, not a primary source.  This report does not contain the original study demonstrating these results.  Either rephrase this sentence to indicate that it is a review or update the citation.

95: “…photoprotective products.” Provide citation.

100: “…oxidative damage”, ageing vs aging

104-107: cite this sentence about the glycoproteins, even if it is the same reference #3 as used in the subsequent sentence, or otherwise indicate that this is part of the same study.  I wanted to know where this data came from.

111: Clinical data is not provided in reference #3.  This sentence dealing with the Skin-Visiometer® also requires citation.

118-119: Significant reference issue.  Reference #5 was performed in lung lymphoblasts, not keratinocytes.  This study did not address IL-8 or TNF-α.

130: “reducing P53 expression which…” this sentence is somewhat unclear.  Recommend breaking it into two sentences: “reducing p53 expression.  P53 is well-known to accumulate…[7]” this also resolves a reference ambiguity.

136: “protective effects of EUK-134 on human keratinocytes” citation needed.

148: inappropriate use of capital letters in “Sugar Biology”

154: “improved on the first day of treatment” and (“in vivo proven” should be changed to “in vivo demonstrated”)

155: proven should be changed to demonstrated

172: inappropriate capital letter usage in Elastin; needs extra term for clarity: “Collagen, elastin, and ECM-degrading enzymes…”

176: “Generated peptides act” subject/verb agreement

192: inappropriate capital letter usage in “Biotechnology”

198: various grammatical.  “Until a decade ago cosmetics were not broadly associated with science”

199: “…uninterested in what the possible…” recommend removing the term what: “…uninterested in the possible…”

201: inappropriate use of capital letters in “Cosmetics” and “Companies”.  Need to rephrase “allergic to science”; recommend “unscientific”

203: inappropriate use of capital letters in “Medical Science and Biotechnology”

204: inappropriate use of apostrophe: “…due to growing consumer demand”

205:  inappropriate use of capital letters in “Science, Medicine, and beauty”

206: “By understanding the basic scientific mechanism” (singular/plural issue)

210: “…is not anymore appealing nor desiring” recommend rephrasing for clarity: “…is no longer appealing nor desirable.”

212: recommend replacing “higher standards” with “high standards”.  Higher is comparative: higher than what?

215: inappropriate use of capital letter in “Cosmetics”

216: inappropriate use of capital letter in “Biotechnology”

218-219: inappropriate use of capital letters in “Cosmetics, Pharmaceutics, and Nutraceutics”

224: recommend rephrasing “Even though the requirements for the efficacy…” the original phrasing is unclear.

240: recommend removing term “perfectly” as we can not make a product 100% completely safe.

241: recommend removing term “all” from “all possible risks”—again, we can not ensure complete safety.

248: “fairness to both parties”

250: “…we assisted to peaks…”  recommend rephrasing to “we greatly increased investment in…”  we did not climb a mountain (ascend vs. assist)

Ref 14: citation issue.  This article was originally published in Personal Care Magazine, Jan 2004 pp 9-14, NOT on ResearchGate!

All the above mistakes were edited.

Reviewer 2 Report

line 111-114 you have to cite the study with visioscan

Author Response

We really appreciated the referee’s suggestions, they were all useful to improve the quality of the manuscript. All referee’s comments have been accepted and revised in the text.

All changes were highlighted in the “rebuttal note” version. Changes in the reference list and numbering in the new text were not highlighted to simplify the reading. Please find the answer to each comment below in red.

Reviewer 2:

Comments and Suggestions for Authors

line 111-114 you have to cite the study with visioscan

Unfortunately this study was not described in the published article Apone et al. 2010 (ref. 6) since the results were obtained after its publication, although the data can be retrieved on line by downloding the active ingredient dossier. Thus we cited the web site as we did in our previous articles published on Cosmetics.  

Reviewer 3 Report

Dear Authors,

I've read your manuscript with huge interest. I'm very glad to see your passion and knowledge about the biotechnology and its use for cosmetic ingredients.

I have few comments, I'm listing below. I'd be very grateful for your answers before the final recommendation to the Editor.

Comment 1:

Line 30-36: It is hard to understand, what timeline you mean in this part. Is it until now that cosmetic industry didn't really had any requirements regarding efficacy? As you wrote later in your article, the legislation about cosmetics have been existing in both US and EU for a long time, not mentioning even legislation for single countries, which varies. Also claiming that mesmerizing advertisement were implied on the most of cosmetic products is hard to understand. Do you mean entire world? Europe? US? How do you know that? Have you analysed many product on the market to claim that? If not, please reformulate to take the "negative expression" down a little. The same about safety: I'm wondering why are you saying that there were only few studies regarding human safety.

Comment 2:

Line 44-45: you wrote "By definition, active ingredients, scientifically proven to be effective on the skin, are called "cosmeceuticals". Well, that is true that A. Kligman coined this term, However, I'd say that this name refers also to final products not only active ingredients. Maybe you can mention that. Later you write that you use this term to describe actives, which is fine, just don't forget that majority of consumers think probably about final products.

Moreover, "cosmeceuticals" refer not only to aged or photoaged skin, but also to other conditions like acne or pigmentation issues.

Comment 3:

Line 100-102: I'd like to see reference(s) to this meaning, just after then or later in this paragraph. You have only reference 3 in line 111, but you say "studies" in pluralis. You also give no reference to a study mentioned in line 111-114.

Comment 4:

Line 141-147: I'd like to see references there, especially about effect of unsaturated fatty acids. And please add reference for desquamation. 

If something has been used for years doesn't need to be good...

Comment 5:

Line 148-155: is reference 9 referring to all information in this paragraph?

Comment 6:

Line 156-164: is reference 10 referring to all information in paragraph, including the clinical test?

Comment 7:

Lines 165-196: This is the entire part called "cell repairing and matrix booster ingredients." However you mention only KTTKS and pal-KTTKS. There is much more other peptides and EMC-boosting ingredients that could be mentioned here. I know that you don't have much article space but I think that readers would expect more than only one peptide in such title and review. You don't need to describe their mode of action in details, but mentioning and references would be very useful, as your article is a review. Of course, you mention two other references, 15 and 16, but you don't give any names.

Comment 8:

Line 198: "Until a decade ago..." - where did you get this date from?

Comment 9:

Line 222: not 40 but 38 years, as EU CD comes from 1976. Please give also references to both EU and US legislations, also in text later.

Comment 10:

Line 233: Please give reference to Colipa guidelines

Author Response

We really appreciated the referee’s suggestions, they were all useful to improve the quality of the manuscript. All referee’s comments have been accepted and revised in the text.

All changes were highlighted in the “rebuttal note” version. Changes in the reference list and numbering in the new text were not highlighted to simplify the reading. Please find the answer to each comment below in red.

Reviewer 3:

Dear Authors, I've read your manuscript with huge interest. I'm very glad to see your passion and knowledge about the biotechnology and its use for cosmetic ingredients.

I have few comments, I'm listing below. I'd be very grateful for your answers before the final recommendation to the Editor.

Comment 1:

Line 30-36: It is hard to understand, what timeline you mean in this part. Is it until now that cosmetic industry didn't really had any requirements regarding efficacy? As you wrote later in your article, the legislation about cosmetics have been existing in both US and EU for a long time, not mentioning even legislation for single countries, which varies. Also claiming that mesmerizing advertisement were implied on the most of cosmetic products is hard to understand. Do you mean entire world? Europe? US? How do you know that? Have you analysed many product on the market to claim that? If not, please reformulate to take the "negative expression" down a little. The same about safety: I'm wondering why are you saying that there were only few studies regarding human safety.

We thank the referee for his/her comments and appreciation. The text was revised accordingly (revised manuscript: lines 27-35).

Comment 2:

Line 44-45: you wrote "By definition, active ingredients, scientifically proven to be effective on the skin, are called "cosmeceuticals". Well, that is true that A. Kligman coined this term, However, I'd say that this name refers also to final products not only active ingredients. Maybe you can mention that. Later you write that you use this term to describe actives, which is fine, just don't forget that majority of consumers think probably about final products.

We thank the referee for his/her comments and appreciation. The text was revised accordingly (revised manuscript: lines 79-85).

Comment 3:

Line 100-102: I'd like to see reference(s) to this meaning, just after then or later in this paragraph. You have only reference 3 in line 111, but you say "studies" in pluralis. You also give no reference to a study mentioned in line 111-114.

Additional references were added (revised manuscript: lines 159-161).

Comment 4:

Line 141-147: I'd like to see references there, especially about effect of unsaturated fatty acids. And please add reference for desquamation.

If something has been used for years doesn't need to be good...

Additional references were added (revised manuscript: lines 200 and 210)

Comment 5:

Line 148-155: is reference 9 referring to all information in this paragraph?

Yes, the reference in line 219 is referring the all information in the paragraph.

Comment 6:

Line 156-164: is reference 10 referring to all information in paragraph, including the clinical test?

Yes, the reference in line 222 is referring the all information in the paragraph.

Comment 7:

Lines 165-196: This is the entire part called "cell repairing and matrix booster ingredients." However you mention only KTTKS and pal-KTTKS. There is much more other peptides and EMC-boosting ingredients that could be mentioned here. I know that you don't have much article space but I think that readers would expect more than only one peptide in such title and review. You don't need to describe their mode of action in details, but mentioning and references would be very useful, as your article is a review. Of course, you mention two other references, 15 and 16, but you don't give any names.

The text was modified accordingly (revised manuscript: lines 255-271)

Comment 8:

Line 198: "Until a decade ago..." - where did you get this date from?

We changed the text as follows: “Until recent times”

Comment 9:

Line 222: not 40 but 38 years, as EU CD comes from 1976. Please give also references to both EU and US legislations, also in text later.

The mistake was corrected and references were added.

Comment 10:

Line 233: Please give reference to Colipa guidelines

Reference 28 regarding Colipa  was added.

Round 2

Reviewer 3 Report

Dear Authors,

Thank you very much for correcting the manuscript. I read it with a great pleasure and I hope to see it in printing soon.

Best regards,

Your reviewer